# Modelling the balance of care: Impact of an evidence-informed policy on a mental health ecosystem

**Nerea Almeda**[1], **Carlos R. Garcia-Alonso**[2], **Mencia R. Gutierrez-Colosia**[1], **Jose A. Salinas-Perez**[2]*, **Alvaro Iruin-Sanz**[3], **Luis Salvador-Carulla**[4]

**1** Department of Psychology, Universidad Loyola Andalucía, Seville, Spain, **2** Department of Quantitative Methods, Universidad Loyola Andalucía, Seville, Spain, **3** Instituto Biodonostia, Red de Salud Mental Extrahospitalaria de Gipuzkoa, Donostia-San Sebastián, Spain, **4** Centre for Mental Health Research, Research School of Population Health, ANU College of Health and Medicine, Australian National University, Canberra, Australia

\* jsalinas@uloyola.es

**Data Availability Statement:** All files are available from the Dryad digital repository (https://doi.org/

## Abstract

Major efforts worldwide have been made to provide balanced Mental Health (MH) care. Any integrated MH ecosystem includes hospital and community-based care, highlighting the role of outpatient care in reducing relapses and readmissions. This study aimed (i) to identify potential expert-based causal relationships between inpatient and outpatient care variables, (ii) to assess them by using statistical procedures, and finally (iii) to assess the potential impact of a specific policy enhancing the MH care balance on real ecosystem performance. Causal relationships (Bayesian network) between inpatient and outpatient care variables were defined by expert knowledge and confirmed by using multivariate linear regression (generalized least squares). Based on the Bayesian network and regression results, a decision support system that combines data envelopment analysis, Monte Carlo simulation and fuzzy inference was used to assess the potential impact of the designed policy. As expected, there were strong statistical relationships between outpatient and inpatient care variables, which preliminarily confirmed their potential and a priori causal nature. The global impact of the proposed policy on the ecosystem was positive in terms of efficiency assessment, stability and entropy. To the best of our knowledge, this is the first study that formalized expert-based causal relationships between inpatient and outpatient care variables. These relationships, structured by a Bayesian network, can be used for designing evidence-informed policies trying to balance MH care provision. By integrating causal models and statistical analysis, decision support systems are useful tools to support evidence-informed planning and decision making, as they allow us to predict the potential impact of specific policies on the ecosystem prior to its real application, reducing the risk and considering the population's needs and scientific findings.

10.5061/dryad.8w9ghx3mp) We have uploaded the dataset to Dryad Digital Repository, and it will be available for the public once the article has been published. In addition, Dryad has provided a temporal link for reviewers so they can access to the dataset. This is the link: https://datadryad.org/stash/share/eds2HxYMKOcALM8zfO7_bZqs5T6vKspmW8rUIOiPKJc Nevertheless, in the text, we should maintain the original DOI for public access because the reviewers link is temporal and only for revision process.

**Funding:** Financial support for this study was provided in part by a grant from the Carlos III Health Institute (PI18/01521), and the Regional Government of Andalusia (PY18-RE-0022), with European Union FEDER teams. The funders had no role in study design, data collection and analysis, decision to publish, or preparation of the manuscript.

**Competing interests:** The authors have declared that no competing interests exist.

## Introduction

The balance of care model is a major driver of the design and monitoring of Mental Health (MH) ecosystems. It was initially proposed as a framework to balance hospital and community integrated care [1]. According to this framework, services should be available depending on the national income level, but it does not provide any practical suggestions regarding the most appropriate number of services, their capacity, workforce characteristics, and other factors.

The incorporation of modelling and scenario assessment in MH care policymaking [2] has facilitated the transition from the balance of care model to actual knowledge-to-action planning. The meta-community model considers a broader range of services, such as social, housing and homelessness services, justice, education and employment [3]. Following this holistic approach, the analysis of the MH balance of care should not be restricted to hospital and community care. From a health ecosystem perspective, the balance-of-care model is intrinsically a systems-based approach. Therefore, it does not intend to reach symmetry between hospital and community services or to compare evidence of one against the other. In contrast, the model aims to find an optimal balance for improving efficiency [4].

There have been major efforts worldwide to promote a better balance in specialized MH care [5]. The World Health Organization Comprehensive Mental Health Action Plan 2013–2020 places a major emphasis on the use of information in developing community care [6]. However, the analysis at the country level is hampered by the ecological effect in multilevel analysis (e.g., the evaluation at the macro level overshadows key variations at the meso level) and requires additional efforts to assess actual resource allocation at the regional and local levels to understand and to monitor the balance of care.

Regardless of the complex relationships between the MH ecosystem and other health or health-related sectors, any integrated and balanced MH ecosystem includes both hospital and community-based care [1]. There are many intuitions about the causal relationships between inpatient and outpatient care, but so far, no clear statistically based evidence has been documented. Outpatient care (aftercare) contributes to reducing relapses and readmissions in inpatient care services [7]. Integrated care from inpatients to outpatients is highly recommended to address service user needs by providing customized individual treatments provided by well-coordinated professional groups [8, 9]. Length of stay and hospitalizations in inpatient care are lower in MH ecosystems that provide a flexible transition between the two types of services [10].

According to the European Commission [11], public policy should enable the best decisions to fulfil population needs efficiently and with high quality standards. The World Psychiatry Association [12] stated that the development of the main balanced MH care model principles is a common challenge worldwide. For this model, any strategy based on enhancing MH outpatient care should consider its potential impact on inpatient care, or vice versa. Therefore, causal relationships between both types of care are assumed.

To enhance MH outpatient care, there are three key elements to consider: the target population, the number of outpatient services, and the number and type of health professionals in these services. The first modifies the demand, and the last two vary the structure of the care provided.

Any real intervention involving these elements will have a real impact on the pre-implementing situation (reality), but a priori, it is impossible to know exactly what that impact can be, the post-implementing situation. Any impact assessment needs to estimate reasonable potential consequences [13] by defining a "reasonable" potential post-implementing situation according to expert-based expectations. In this potential underlying causal model, variable relationships should be defined by experts because they can identify potential "causes" and

"effects", as well as different "causal levels" that can define which variables can be "effects" and then "causes" of others. A causal model can be represented by a graph by linking "causes" and "effects" by, usually unidirectional, arrows. This structure is relatively easy to understand and is a powerful tool to guide statistical analysis. In this study, a Bayesian network (Direct Acyclic Graph) is selected to design the causal model.

In the real word causes are usually easy to manage (they are decisions), but their potential impacts on the ecosystem are not easy to predict (they are out of managerial control). For example, if the number of psychiatrists in an outpatient facility is increased, then the number of visits in this facility should increase, but by how many visits? Consequently, the number of days of stay in inpatient facilities should decrease, but by how many days?

Evidence-informed decision-making requires expert-based validation of underlying causal relationships [14]. This requires that decision-makers participate in an iterative process for incorporating new information for modelling. Then, emerging behaviours highlighted by statistical techniques such as regression, factor analysis, structural equations, etc. [15] must be checked again by expert panels to avoid spurious results, and their knowledge must be integrated again into the analysis following an iterative process.

In this paper, the structure of a potential causal model linking outpatient and inpatient care variables was first designed based on expert knowledge. Data from the MH ecosystem of Gipuzkoa were then used [16] to identify a potential statistical-base structure for the variable relationships using generalized regression with product unit and exponential bases. Considering the low number of observations and the existence of outliers, the use of factor and/or structural equation analysis is compromised. Therefore, this study must be considered exploratory. Previous works [17] have shown that it is possible to estimate the potential impact of a policy on MH ecosystem performance once expert knowledge is adequately formalized. To assess performance, relative technical efficiency (RTE) [18], statistical stability and Shannon's entropy [19] were ultimately used.

The objectives of this paper were (i) to design a formal causal model (Bayesian network prototype) based on expert knowledge and the balance of care model linking inpatient and outpatient MH care, (ii) to identify, if they exist, statistical relationships between selected variables, and finally (iii) to assess the potential impact of a specific policy enhancing the inpatient-outpatient MH care balance on real ecosystem performance based on (i) and (ii). Everything was evaluated in a low-observation, 13 MH catchment areas, and uncertain environment.

## Methods

### Terminology

The Description and Evaluation of Care Services and Directories for Long Term Care (DESDE-LTC) tool was used for standardized care provision [20–22]. DESDE-LTC is a classification system for coding care units and service availability, allowing international comparisons across different jurisdictions. In this study, Care Teams or Basic Stable Inputs of Care provide the following main types of care: Residential ("R" DESDE-LTC code) and Outpatient care ("O" DESDE-LTC code).

In addition, an international glossary of terms [23] were used to overcome terminology and commensurability problems in MH ecosystem research.

### Bayesian network

Bayesian network prototypes were designed by formalizing expert knowledge from the Department of Health of Gipuzkoa (Basque Country, Spain). This knowledge was registered and saved in successive technical meetings throughout 5 years following the Expert-based

Cooperative Analysis methodology [24]. The final model was improved, and their relationships statistically confirmed by regression analysis.

## Sample and variables

Gipuzkoa has a population of 640,635 adults older than 17 years of age in 2015. It is one of the three historic territories of Basque Country autonomous community in Spain. The Department of Health in each historic territory has total governance capacity and centralizes healthcare management and provision [25]. The MH ecosystem of Gipuzkoa is structured in 13 catchment areas, which are considered the decision-making units (DMUs) for policy assessment. Each catchment area of Gipuzkoa corresponds to a community MH centre. A single acute MH hospital unit provides care to all the DMUs [16]. The 13 catchment areas are: Alto Deba-Arrasate, Amara, Andoain, Azpeitia, Beasain, Eguia, Eibar, Irun, Ondarreta, Renteria, Tolosa, Zarautz and Zumarraga.

In this study, 85 variables were used to describe MH care provision and to assess ecosystem performance by using RTE scores. Seventy-five variables were identified as resources used by the DMUs to provide MH care, which were classified as inputs in Group C: quality of care [23]. Ten variables were classified as outcomes (outputs) obtained by using the inputs, classified in Group B: service utilization. However, only a few of these variables are considered by experts for designing potential causal relationships between outpatient and residential MH care. Data set is available at the Dryad digital repository (https://doi.org/10.5061/dryad.8w9ghx3mp).

## Scenarios

A scenario is a combination of potential causally-related variables (inputs and/or outputs) that provides a specific perspective of the MH ecosystem status and its evolution over time. The ecosystem global status can be assessed by the mathematical integration of the results obtained from all the scenarios. In this study, experts in MH designed 15 scenarios related to the main types of care: residential, day and outpatient care. It is worthy to highlight that those experts who participated in this panel were senior managers, psychiatrists, clinical psychologists, nurses and researchers, who had experience in planning and managing MH services and worked in the MH Network of Gipuzkoa and Bizkaia (another historic territory in the Basque Country, Spain). Altogether 15 experts participated in designing both the scenarios and the potential policy to evaluate.

## Techniques

The first step was to design a set of expert-driven Bayesian network prototypes. These graph-based structures define potential causal relationships between variables based on expert knowledge. Here the experts have to decide which variable is a "cause" and which variable is an "effect" according to his/her knowledge. The resulting diagram (usually a Direct Acyclic Graph) represents an individual o collective (more appropriate) proposal that can, or not, be statistically confirmed by data. Without this underlying model it is very difficult to carry out any exploratory and confirmatory analysis or, even worse, to draw conclusions from blind regression or classification techniques. Proposed graphs must be checked iteratively, Expert-based Cooperative Analysis [24], by the panel because variable selection and dependency levels are usually difficult to define.

As the number of observations was very low and knowing the existence of potential outliers, it was not feasible to run an exploratory factorial analysis or to check the models using structural equations. For this situation, standard regression was selected to initially explore the corresponding statistical relationships, which are not linear.

Once a Bayesian network prototype was designed, generalized least squares [26] on exponential [1.1] and product unit [1.2] base functions were used to highlight the potential emerging behaviours identified by the experts.

$$y = \alpha e^{\beta x_1 x_2 \dots x_n} \qquad [1.1]$$

$$y = \alpha x_1^{\beta_1} x_2^{\beta_2} \dots x_n^{\beta_n} \qquad [1.2]$$

where $y$ is the dependent variable (potential effect), $x_i$, $i = 1, 2, \dots, n$ represents the independent variables (potential causes), and $\alpha$, $\beta$ and $\beta_j$, $j = 1, 2, \dots, n$ are the parameters that define the potential relationship. These structures are very flexible, linearizable, and, when the number of independent variables is relatively low, easy to explain. The specific form [1.1 or 1.2] for the basis function for each potential causal relationship and the best values for $\alpha$, $\beta$ and $\beta_j$ were iteratively improved and calculated by checking the statistical error (minimizing).

The proposed Bayesian network defines the independent and dependent variables of each potential causal relationship. These relationships were iteratively modified and improved according to the regression results. Expert panel defined the first model and checked both the statistical results and the proposed changes in the Bayesian network until they agreed with the results.

The best Bayesian networks and their corresponding regression models were used to estimate the statistical ranges or confidence intervals for the dependent variable (potential effect, consequence, or output) given the independent variable values. These ranges, adjusted to delimited statistical distributions (triangular, trapezoidal and gamma), were managed by a Monte Carlo simulation engine for intervention assessment.

A computer-based Decision Support System (DSS) [17] was used to assess the status of ecosystem performance: RTE, statistical stability and entropy. For RTE, data envelopment analysis with variable returns to scale was used considering both the input orientation, which tries to minimize the inputs while keeping the outputs constant; and the output orientation, which tries to maximize the outputs while keeping the inputs constant. The RTE scores were within [0, 1]: 0 denotes complete inefficiency, and 1 shows complete efficiency.

A Monte Carlo simulation engine was designed to include randomness in the RTE analysis [17]. All the original data were transformed into a statistical distribution according to the confidence interval calculated by the best regression models. The uniform distribution was finally selected, where the left-right limits were established by the respective confidence intervals.

Stability assesses whether small variable value changes can significantly vary the RTE scores, in which, 0% is completely unstable and 100% is completely stable. Shannon's entropy was used to analyse the homogeneity of the ecosystem management from completely homogeneous, i.e., if all the catchment areas are managed exactly in the same way, the entropy will be 0%, to completely heterogeneous, if each has its own method of management the entropy will be 100%.

## The policy

Decision-makers, senior managers of the MH System of Gipuzkoa, wanted to reinforce MH outpatient care to improve the inpatient/outpatient balance of care. The policy was structured by the following interventions:

- The number of psychiatrists on outpatient care teams was balanced among catchment areas according to the expected population needs.

- New full-time psychiatrists were hired.

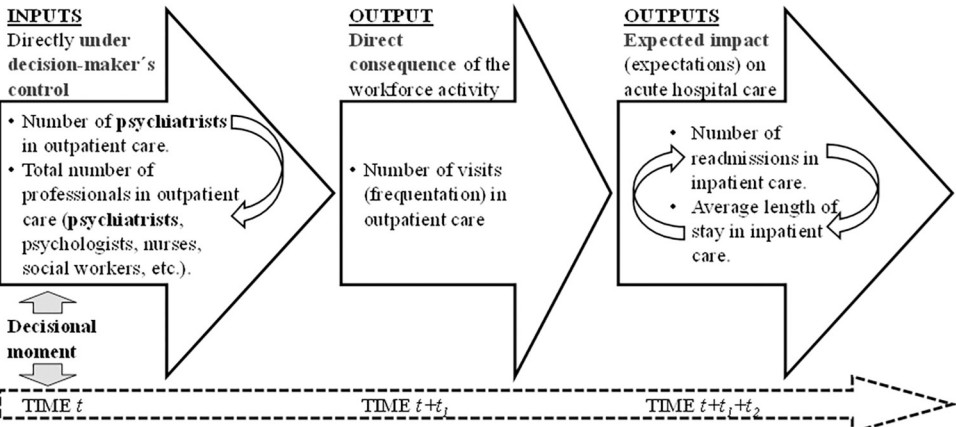

**Fig 1. Sequence of the decisional consequences (causality) in the ecosystem and variables involved.**

Ten out of 15 scenarios were involved in the policy. The modified variables were (Fig 1 (time $t$)):

- Number of psychiatrists in outpatient care, and

- Total number of health professionals in outpatient care.

These modifications had causal impacts on the following variables (Fig 1 (time $t+t_1$), with $t_1$ being small):

- Frequentation, number of visits, in community mental health centres.

    Consequently, there was an expected impact on:

- Average number of readmissions, number of readmissions divided by the number of discharges, and

- Length of stay, days of stay divided by the number of discharges, (see Fig 1, time $t+t_1+t_2$, $t_2$ medium-long).

Decision-makers expected an increase in the number of visits in outpatient care facilities, specifically the non-mobile and non-acute services with codes O8-O10 according to DES-DE-LTC, and a decrease in both the length of stay and the number of readmissions in inpatient care in the catchment areas, specifically the services that provide acute hospital medium intensity care with code R2 according to DESDE-LTC.

## Decisions (inputs, time t)

- Reassignment of a half full-time-equivalent psychiatrist from the Tolosa community MH centre to the Azpeitia centre (outpatient care).

- Hiring of a full-time equivalent psychiatrist for the Andoain community MH centre (outpatient care).

The objective is to rebalance care provision in the pre-implementing policy situation (reality) by increasing the number of high qualified professionals in outpatient care.

**Table 1. Basic statistics for original data.**

| | Population (inhabitants) | Frequentation (visits) | Length of stay (days) | Discharges (users) | Readmissions (users) | Outpatient services per 100,000 | Psychiatrists per 100,000 | Psychologists per 100,000 | Nurses per 100,000 | Total number of professionals per 100,000 |
|---|---|---|---|---|---|---|---|---|---|---|
| Average | 49,279.62 | 14,030.85 | 1,112.69 | 75.23 | 6.23 | 2.44 | 5.93 | 2.11 | 3.93 | 13.89 |
| Standard deviation | 20,942.74 | 7,414.42 | 620.42 | 43.32 | 3.92 | 1.23 | 1.29 | 1.24 | 0.90 | 2.65 |
| Variation coefficient (%) | 42.50 | 52.84 | 55.76 | 57.59 | 62.90 | 50.15 | 21.76 | 58.92 | 22.87 | 19.06 |
| Minimum | 21,593 | 5,532 | 392 | 28 | 1 | 1.25 | 4.31 | 0 | 2.47 | 10.91 |
| Maximum | 78,400 | 27,205 | 2.058 | 147 | 15 | 4.53 | 9.06 | 4.08 | 5.51 | 21.11 |

## Results

Basic statistics for original data, pre-implementing situation, are shown in Table 1.

### Causal relationships identified by the experts

The potential causal relationships identified by the experts were as follows:

- The number or visit frequency in outpatient centres (O8 to O10 DESDE-LTC codes) depends on the number of health professionals.

- The length of stay in inpatient units (R2 DESDE-LTC code) in each area depends on the number of outpatient facilities in the area and the numbers of stays and health professionals in outpatient facilities.

- The number of readmissions in acute hospital units depends on the number of outpatient centres in the area and the numbers of stays and health professionals in outpatient services.

The causal nature of the phenomenon was identified by the decision-makers in Gipuzkoa, and then relationships were checked by generalized linear regression once the product unit functions [1.1 and/or 1.2] were appropriately linearised. Both information sources formed an evidence-informed Bayesian network (Fig 2). For a specific population $Pop_i$, $i = 1, 2, . . .,13$, the total number of professionals in outpatient care $TNProff_i$, $i = 1, 2, . . .,13$ is the key variable due to the specific characteristics of this type of care, which is provided by well-defined care teams.

### Causal relationships based on data (outputs, times: t+t1 and t+t1+t2)

**Frequentation (visits) in outpatient care (time t+t1).** In Gipuzkoa, a strong exponential relationship was found between population $Pop_i$, $i = 1, 2, . . .,13$, which is defined as number of adults older than 17 years old divided by 1,000, and the total number of professionals in outpatient care $TNProff_i$, $i = 1, 2, . . .,13$ [2] (significance level 0.05, $R^2 = 0.8794$, F = 130.22).

$$TNProff_i = 2.381 \times e^{0.0196 \times Pop_i} \qquad [2]$$

The confidence intervals were [1.932, 2.9343] for $\alpha = 2.3281$ and [0.0156, 0.0235] for $\beta = 0.0196$.

A strong exponential relationship (significance level 0.05, $R^2 = 0.9221$, F = 130.22) among frequentation (number of visits in community mental health centres in thousands) and the corresponding multiplication between the target population by the number of professionals providing outpatient care [3] (the resulting multiplication can be considered a composite

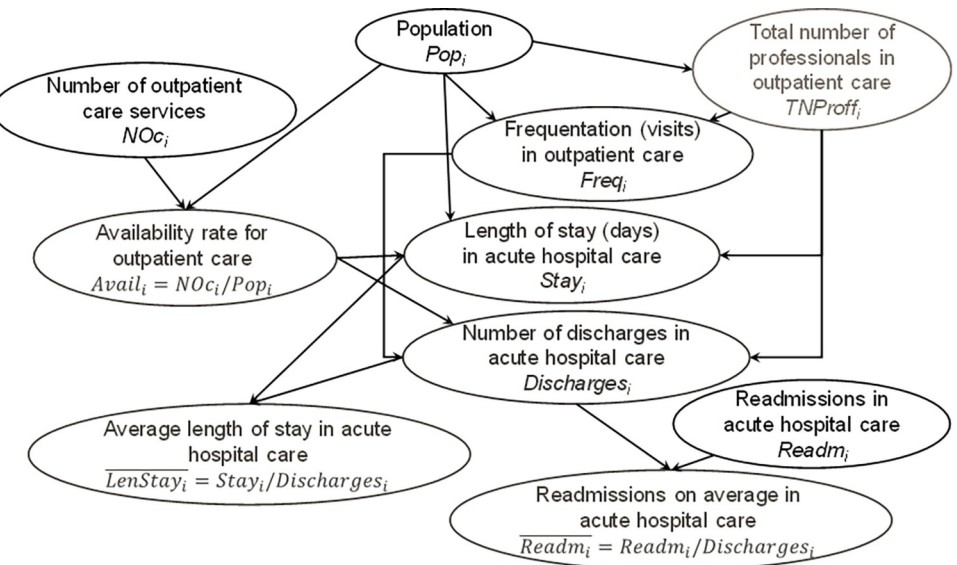

**Fig 2. Final evidence-informed Bayesian network based on expert knowledge and data.**

indicator represented in Fig 2) was also found:

$$Freq_i = 5.7767 \times e^{0.0196 \times Pop_i \times TNProff_i} \qquad [3]$$

where $Freq_i$ is the frequentation (in thousands) for the $i^{th}$ catchment area ($i = 1, 2, \ldots, 13$). The confidence intervals were [5.0079, 6.6634] for $\alpha = 5.7767$ and [0.0165, 0.0227] for $\beta = 0.0196$ (Fig 3).

**Length of stay in inpatient care (time t+t1+t2).** Between length of stay (days) in inpatient care and $Pop_i$ multiplied by $TNProff_i$, there was another strong exponential relationship

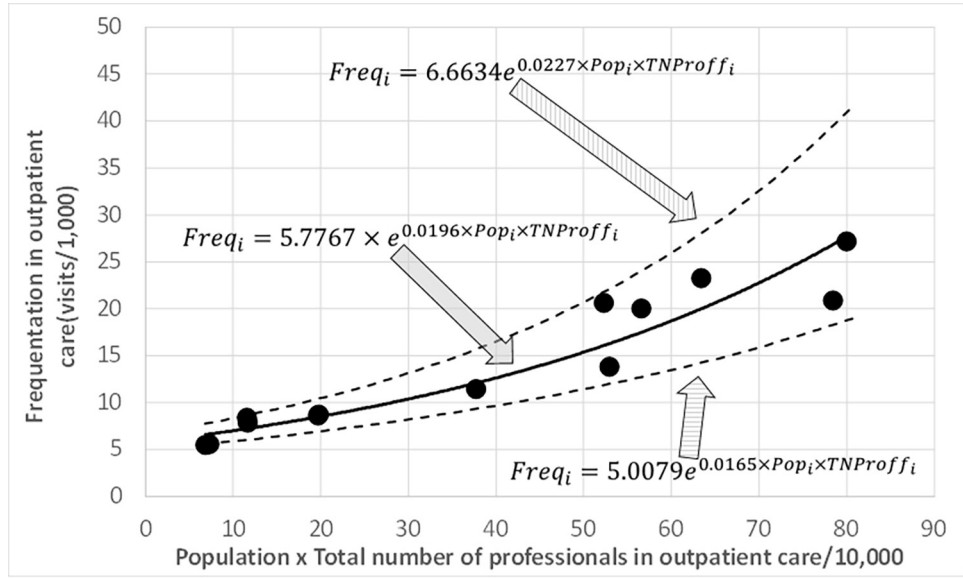

**Fig 3. Relationship between $Freq_i$ and $Pop_i \times TNProff_i$ (significance level 0.05).** Confidence intervals in dashed lines.

[5] (significance level 0.05, $R^2$ = 0.8206, F = 60.65) considering the selected catchment areas:

$$Stay_i = 433.39e^{0.0204 \times Pop_i \times TNProff_i} \qquad [4]$$

where $Stay_i$ is the length of stay (days) in inpatient care of the $i^{th}$ catchment area ($i$ = 1, 2, ..., 13). The confidence intervals were [382.98, 587.14] for $\alpha$ = 433.39 and [0.0142, 0.0236] for $\beta$ = 0.0204. This statistical relationship was complimentary to [6] (significance level 0.05, $R^2$ = 0.776, F = 38.1):

$$Stay_i = 2,269.6 \times Avail_i^{-1.1154} \qquad [5]$$

where $Avail_i$ is the availability of outpatient care services (services per 100,000 inhabitants, age older than 17 years) in the $i^{th}$ catchment area ($i$ = 1, 2, ..., 13). The confidence intervals were [1,689.32, 3,049.23] for $\alpha$ = 2,269.6 and [-1.44, -0.7909] for $\beta$ = -1.1154. This relationship supported one of the fundamental principles of the community MH model: the greater $Avail_i$ was, the lower the value of $Stay_i$ (Fig 3).

Fig 3. Relationship between $Avail_i$ and $Stay_i$.

**Number of readmissions in inpatient care (time t+t1+t2).** There was no relationship between the number of readmissions in inpatient care facilities and the variables related to population and/or outpatient care services. Readmission is a phenomenon that involves relatively few users with likely chaotic behaviour; thus, it is very difficult to find a statistical model that can be used to link readmissions in inpatient care with outpatient care variables.

**Number of discharges in inpatient care (time t+t1+t2).** There was a relationship between the number of discharges in inpatient care and $Freq_i$ multiplied by $TNProff_i$ [7] (significance level 0.05, $R^2$ = 0.8092, F = 46.66):

$$Discharges_i = 5.372 \times (Freq_i \times TNProff_i)^{0.5697} \qquad [6]$$

where $Discharges_i$ is the number of discharges in the inpatient care of the $i^{th}$ catchment area ($i$ = 1, 2, ..., 13). The confidence intervals were [2.764, 10.441] for $\alpha$ = 5.372 and [0.4199, 0.7194] for $\beta$ = 0.5697 ($Freq_i \times TNProff_i$ divided by 1,000). In addition, there was another potential relationship between $Discharges_i$ and $Avail_i$ [8] (significance level 0.05, $R^2$ = 0.7899, F = 41.35):

$$Discharges_i = 155.49 \times Avail_i^{-1.141} \qquad [7]$$

The confidence intervals were [116.35, 207.79] for $\alpha$ = 155.49 and [-1.4597, -0.8224] for $\beta$ = -1.141.

*Ecosystem performance assessment*. The number of simulations was established as 500 (statistical error lower than 2.5%).

## Potential impact on the MH system of Gipuzkoa

**Input-oriented results.** The results showed that the average global RTE ($\overline{RTE}$) increased (Table 2) by 3.38% in the postintervention situation (from 0.83 to 0.86), being both scores very high. The probability of having an efficiency score greater than 0.75 also increased by 4.85% (from 0.76 to 0.8). Therefore, the policy enhances the performance of the global MH ecosystem.

Pre- and postintervention stability indicators highlighted that the MH ecosystem of Gipuzkoa was highly unstable (Table 2). Nevertheless, the stability increased by 24.47% (from 23.46% to 29.2%). Shannon entropy scores showed that ecosystem management was more homogeneous after the policy, although it was still adjusted to territorial characteristics (from 76.69% to 71.18%).

**Table 2. Input-oriented results, variation pre-post (%) in brackets.** In bold, the catchment areas directly involved with the decisional process.

| Areas | Relative technical efficiency (RTE) on average [1] | | Probability of having an RTE score greater than 0.75 [1] | | RTE error | | Stability of the ecosystem [4] (%) | | Shannon's entropy (%) [5] | |
|---|---|---|---|---|---|---|---|---|---|---|
| | Pre [2] | Post [3] | Pre | Post | Pre | Post | Pre | Post | Pre | Post |
| Global (Gipuzkoa MH ecosystem) [6] | 0.83 [7] | 0.86 (3.38) | 0.76 | 0.80 (4.85) | 0.0016 | 0.0019 (22.26) | 23.46 | 29.20 (24.47) | 76.69 | 71.18 (-7.18) |
| Alto Deba-Arrasate | 0.82 | 0.80 (-2.20) | 0.69 | 0.60 (-12.18) | 0.0068 | 0.0100 (47.02) | 28.71 | 22.93 (-20.11) | 66.07 | 74.20 (12.30) |
| Amara | 0.82 | 0.86 (5.01) | 0.63 | 0.77 (22.36) | 0.0042 | 0.0050 (19.74) | 29.71 | 36.12 (21.58) | 65.91 | 64.01 (-2.88) |
| **Andoain** | **0.95** | **0.96 (1.61)** | **0.95** | **0.90 (-4.67)** | **0.0014** | **0.0008 (-41.37)** | **51.17** | **56.26 (9.96)** | **47.55** | **42.46 (-10.71)** |
| **Azpeitia** | **0.85** | **0.91 (7.79)** | **0.83** | **0.91 (10.01)** | **0.0049** | **0.0019 (-60.76)** | **27.68** | **53.56 (93.50)** | **67.57** | **56.39 (-16.54)** |
| Beasain | 0.89 | 0.87 (-2.55) | 0.90 | 0.84 (-6.73) | 0.0021 | 0.0034 (62.42) | 40.97 | 40.02 (-2.32) | 58.53 | 67.89 (16.00) |
| Eguia | 0.74 | 0.83 (12.75) | 0.51 | 0.74 (44.78) | 0.0063 | 0.0031 (-51.33) | 27.98 | 28.33 (1.27) | 78.42 | 66.24 (-15.54) |
| Eibar | 0.77 | 0.78 (0.80) | 0.66 | 0.64 (-2.69) | 0.0042 | 0.0035 (-17.78) | 19.23 | 30.89 (60.63) | 78.26 | 73.29 (-6.34) |
| Irun | 0.73 | 0.78 (8.01) | 0.54 | 0.63 (16.75) | 0.0078 | 0.0049 (-37.66) | 26.22 | 29.38 (12.05) | 79.20 | 71.97 (-9.12) |
| Ondarreta | 0.81 | 0.83 (3.28) | 0.64 | 0.56 (-11.55) | 0.0056 | 0.0069 (22.68) | 30.90 | 29.97 (-3.03) | 74.22 | 57.82 (-22.10) |
| Renteria | 0.82 | 0.85 (3.82) | 0.73 | 0.80 (9.95) | 0.0049 | 0.0070 (41.56) | 32.24 | 36.93 (14.53) | 72.13 | 63.92 (-11.38) |
| **Tolosa** | **0.87** | **0.89 (1.58)** | **0.87** | **0.83 (-3.70)** | **0.0023** | **0.0055 (141.38)** | **28.27** | **32.11 (13.58)** | **72.20** | **70.60 (-2.21)** |
| Zarautz | 0.86 | 0.88 (2.19) | 0.83 | 0.81 (-2.36) | 0.0026 | 0.0028 (10.81) | 37.05 | 44.25 (19.43) | 68.42 | 64.28 (-6.05) |
| Zumarraga | 0.86 | 0.90 (3.93) | 0.87 | 0.75 (-14.42) | 0.0035 | 0.0046 (32.49) | 28.43 | 39.09 (37.51) | 64.22 | 62.80 (-2.20) |

(1) $RTE \in [0, 1]$ (0: DMU completely inefficient, 1: DMU completely efficient); 500 experiments.

(2) Pre: Pre-Intervention.

(3) Post: Post-Intervention.

(4) $Stability \in [0, 100]$ (0: minimum stability–small data changes can result in very large RTE changes; 100 maximum stability–data changes do not modify RTE).

(5) *Shannon´s entropy* is calculated as a percentage of the feasible maximum estimated by the frequency analysis. $Entropy \in [0, 100]$ (0: minimum–the ecosystem has a very homogeneous management, 100: maximum–the ecosystem has a very heterogeneous management).

(6) Fifteen scenarios (meaningful variable combinations (500 experiments each); 13 catchment areas.

(7) The results have been rounded to the most appropriate number of decimal places.

**Output-oriented results.** After the interventions, the $\overline{RTE}$ remained almost constant (-0.14%) at $\overline{RTE} = 0.83$ (Table 3). The probability of having an $\overline{RTE}$ greater than 0.75 was also almost constant (-0.89%). The stability decreased slightly (-0.54%), from 13.89 to 13.81, while Shannon's entropy was constant (0.03%), from 73.94% to 73.96%.

## Impact on the catchment areas

**Input-oriented results.** In all the catchment areas directly involved by the policy, $\overline{RTE}$ increased (Table 2): in Andoain (from 0.95 to 0.96), in Azpeitia (from 0.85 to 0.91) and in Tolosa (from 0.87 to 0.89). In the other areas, Eguia experienced the highest efficiency increase (12.75%), while Beasain (-2.55%) and Alto Deba-Arrasate (-2.20%) suffered a negative impact.

**Table 3. Output-oriented results, variation pre-post (%) in brackets.** In bold, the catchment areas directly involved with the decisional process.

| Areas | Relative technical efficiency (RTE) on average [1] | | Probability of having an RTE score greater than 0.75[1] | | RTE error | | Stability of the ecosystem [4] (%) | | Shannon's entropy (%) [5] | |
|---|---|---|---|---|---|---|---|---|---|---|
| | Pre [2] | Post [3] | Pre | Post | Pre | Post | Pre [2] | Post [3] | Pre | Post |
| Global (Gipuzkoa MH ecosystem) [6] | 0.83[7] | 0.83 (-0.14) | 0.77 | 0.77 (-0.89) | 0.0014 | 0.0010 (-31.88) | 13.89 | 13.81 (-0.54) | 73.94 | 73.96 (0.03) |
| Alto Deba-Arrasate | 0.80 | 0.80 (-0.32) | 0.61 | 0.63 (1.80) | 0.0036 | 0.0069 (92.66) | 12.17 | 12.14 (-0.21) | 77.27 | 77.65 (0.49) |
| Amara | 0.84 | 0.84 (0.26) | 0.78 | 0.78 (0.45) | 0.0046 | 0.0032 (-30.73) | 17.48 | 17.58 (0.54) | 71.06 | 70.46 (-0.84) |
| **Andoain** | **0.95** | **0.96 (0.22)** | **0.91** | **0.91 (-0.04)** | **0.0007** | **0.0015 (103.25)** | **25.34** | **25.39 (0.18)** | **43.66** | **43.34 (-0.72)** |
| **Azpeitia** | **0.87** | **0.87 (0.21)** | **0.81** | **0.81 (-0.05)** | **0.0040** | **0.0034 (-14.01)** | **18.08** | **17.96 (-0.69)** | **68.83** | **66.89 (-2.83)** |
| Beasain | 0.90 | 0.89 (-1.07) | 0.88 | 0.85 (-3.50) | 0.0031 | 0.0021 (-30.89) | 19.38 | 18.85 (-2.74) | 59.94 | 63.29 (5.59) |
| Eguia | 0.79 | 0.79 (0.36) | 0.74 | 0.73 (-0.76) | 0.0082 | 0.0054 (-33.62) | 8.03 | 8.03 (0.00) | 65.44 | 64.84 (-0.91) |
| Eibar | 0.77 | 0.76 (-1.03) | 0.55 | 0.54 (-2.47) | 0.0053 | 0.0043 (-17.90) | 11.05 | 11.14 (0.81) | 68.52 | 67.69 (-1.21) |
| Irun | 0.76 | 0.76 (0.33) | 0.58 | 0.56 (-4.28) | 0.0105 | 0.0116 (9.60) | 9.34 | 9.34 (0.00) | 77.95 | 76.78 (-1.51) |
| Ondarreta | 0.76 | 0.75 (-1.20) | 0.56 | 0.52 (-6.46) | 0.0087 | 0.0032 (-63.38) | 9.00 | 9.00 (0.06) | 74.02 | 73.84 (-0.24) |
| Renteria | 0.84 | 0.83 (-0.35) | 0.72 | 0.71 (-1.81) | 0.0128 | 0.0091 (-28.95) | 13.51 | 13.51 (0.00) | 63.04 | 64.24 (1.91) |
| **Tolosa** | **0.84** | **0.85 (1.14)** | **0.75** | **0.79 (5.93)** | **0.0039** | **0.0033 (-14.02)** | **15.30** | **19.69 (28.70)** | **72.25** | **71.06 (-1.66)** |
| Zarautz | 0.85 | 0.86 (0.05) | 0.76 | 0.76 (0.37) | 0.0059 | 0.0020 (-66.96) | 16.22 | 16.26 (0.25) | 69.19 | 68.81 (-0.55) |
| Zumarraga | 0.87 | 0.86 (-0.56) | 0.66 | 0.64 (-3.27) | 0.0086 | 0.0079 (-8.75) | 15.65 | 14.10 (-9.90) | 68.23 | 67.09 (-1.68) |

(1) *RTE*∈[0, 1] (0: DMU completely inefficient, 1: DMU completely efficient); 500 experiments.

(2) Pre: Pre-Intervention.

(3) Post: Post-Intervention.

(4) *Stability*∈[0. 100] (0: minimum stability–small data changes can result in very large RTE changes; 100 maximum stability–data changes do not modify RTE).

(5) *Shannon´s entropy* is calculated as a percentage of the feasible maximum estimated by the frequency analysis. *Entropy*∈[0, 100] (0: minimum–the ecosystem has a very homogeneous management, 100: maximum–the ecosystem has a very heterogeneous management).

(6) Fifteen scenarios (meaningful variable combinations, 500 experiments each); 13 catchment areas.

(7) The results have been rounded to the most appropriate number of decimal places.

For the probability of having an RTE score greater than 0.75 (Table 2), Andoain decreased (from 0.95 to 0.9), Azpeitia increased (from 0.83 to 0.91) and Tolosa decreased (from 0.87 to 0.83). In the other areas, Eguia (44.78%) and Amara (22,37%) increased significantly, while Zumarraga (-14.42%) and Alto Deba-Arrasate (-12.18%) showed negative impacts.

Andoain (from 51.17 to 56.26%), Azpeitia (from 27,68 to 53.56%) and Tolosa (from 28,27 to 32.11%) increased their stabilities (Table 1). Other areas had relevant increases, such as Eibar (60,63%) and Zumarraga (37,51%), whereas Alto Deba-Arrasate (-20.21%) had a relevant decrease.

Shannon´s entropy was very high in all areas (Table 2), indicating that ecosystem management was highly customized. Andoain (from 47.55 to 42.46%), Azpeitia (from (7) 67.57 to

56.39%) and Tolosa (from 72.2 to 70.6%) decreased their scores significantly (more heterogeneous management). For this indicator, Ondarreta (-22.10%) and Eguia (-15.54%) decreased significantly (more homogeneous management), while Beasain (16%) and Alto Deba-Arrasate increased significantly (12.30%).

**Output-oriented results.**  In policy-involved areas, $\overline{RTE}$ remained constant in Andoain (0.96) and Azpeitia (0.87), but Tolosa had an increase, from 0.84 to 0.85 (Table 3). None of the other areas reached a relevant increase, and only Ondarreta had a small decrease (-1.2%).

Regarding the probability of having an RTE score greater than 0.75 (Table 3), only Tolosa had a small increase, from 0.75 to 0.79; in Andoain and Azpeitia, the score remained approximately constant. In the other areas, Alto Deba-Arrasate increased slightly (1.8%), but Ondarreta (-6.46%) and Irun (-4.28%) had negative impacts.

The stability in Andoain and Azpeitia remained approximately constant (Table 3), and Tolosa showed a relevant increase from 15.3 to 19.69%. No other areas had an increase; conversely, Zumarraga (-9.9%) showed a negative impact.

Ecosystem management was highly customized (Table 3). Andoain, Azpeitia and Tolosa decreased their entropies, but only the decrease in the second can be considered significant, from 68.83 to 66.89. In the remaining areas, Zumarraga (-1.68%) decreased slightly, while Beasain (5.59%) highlighted a significant increase, which shows a more heterogeneous management.

## Discussion

To the best of our knowledge, this is the first study that has identified, assessed and formalized the causal relationships between descriptors for inpatient and outpatient care for real-world evidence-informed planning. This model has been used to assess the potential impact of a specific policy designed to balance inpatient and community-based care provision, by using the Bayesian network integrating the expert knowledge and statistical approach.

Decision-makers usually have a defined strategy, but it is difficult for them to estimate its potential impact. DSS can be used for that purpose but require the integration of expert knowledge [27]. To support the decision-making process, two elements stand out: the first focuses on evidence assessment, and the second refers to the participation of decision-makers [28]. In this research, senior decision-makers of the Gipuzkoa MH system identified the potential causal relationships, defined the policy and actively participated in the process. It is worthy to note that they consider the DSS a useful tool for supporting and guiding policymaking, nevertheless, the analysis of the concordance between the reality (pre-implementing situation) and the predictions (potential post-implementing situation) is critical to validate the model.

It is worthy to note that he relationship between outpatient care provision and general inpatient care improvement is acknowledged [29, 30]. In the selected policy developed in this research, the effects on the visits in outpatient care services occur in weeks or months, but those corresponding to inpatient care services in terms of length of stay, discharges and readmissions are expected to have a significant delay, taking place in months or years.

Results obtained by Weiss et al. (2012) [31] showed the importance of integrating emergency departments in a MH system that provide aftercare to promote the transition of patients from inpatient care. Findings from Hansagi et al. (2000) [32] evidenced that users who visited actively emergency department also heavily use general health care services. These facts result interesting taking into consideration the health crisis caused by Covid-19 pandemic. The change of paradigm that we are living has forced the development of new ways of treatment in order to avoid physical contact and spread the virus. In this sense, telepsychiatry has played a crucial role in providing MH care, which has rapidly evolved because of the pandemic [33].

This type of care presents advantages such as it is easier to provide continuity of care, to triage emergency patients and to reduce economic costs in providing outpatient care [34]. Although the pandemic has triggered the implementation of telepsychiatry, previous studies focused on the future of psychiatry, highlighting the importance of technology to transform the traditional MH care provision [35–37].

Policies or interventions not only have a direct impact on the target catchment areas but also have indirect effects on the rest because of an input/output rebalance. In the areas where the impact was negative, additional adjustments could be needed to improve their performance.

When $\overline{RTE}$ is high (greater than 0.8), it is very difficult to design a policy to improve the performance. In Gipuzkoa, the stability was low or very low. In this situation, only small organizational interventions with a strict monitoring process should be considered by cautious decision-makers. The results for entropy showed a slight decrease in the input orientation; therefore, the management was slightly more homogeneous. Previous work on the assessment of a policy intervention in Bizkaia (Basque Country) showed that, after shifting workforce capacity from an outpatient to day care service, the global performance of the MH system increased [17].

Future lines of work could be focused on improving outputs used for assessing RTE by including telephonic sessions in outpatient care, which have been increased after the covid pandemic.

## Conclusions

The objectives of this article were: (i) to design a formal causal model (Bayesian network prototype) based on expert knowledge and the balance of care model linking inpatient and outpatient MH care, (ii) to identify, if they exist, statistical relationships between selected variables, and finally (iii) to assess the potential impact of a specific policy enhancing the inpatient-outpatient MH care balance on the MH system of Gipuzkoa.

It is possible to design expert-based Bayesian networks independently from the number of observations. This approach can be more realistic for causality formalization (15), but it is not enough. Data will be necessary to confirm, if they exist, the mathematical structure of the relationship or emerging behaviour. This process must be designed in an iterative way to explore different algebraic structures (linear, product units and/or exponential) automatically to find the best resulting model (regression analysis). Therefore, statistical analysis is used to discover hidden relationships that are difficult to highlight by experts. These new relationships must be confirmed by the experts and then included in the Bayesian network, that is, the result of a combination of expert knowledge and statistical-based results. Results have shown that is possible to identify the causal relationships according inpatient and outpatient care according to the expert knowledge and from a statistical point of view. Nevertheless, it is important to note that the resulting Bayesian network is specific for the ecosystem under study. Each new situation has to be analysed according the existing, local, knowledge but the methodology is completely replicable.

Once the potential causal relationships, and their levels, are confirmed for the situation, results must be integrated for the assessment of the policy impact. This analysis have to understand that reality is not under certainty but completely uncertain. This means that each specific data (for example, the frequentation of Tolosa -catchment area) is a statistical distribution, the problem is which one. The best statistical models, specifically their confidence intervals, are used to identify the statistical distributions which are evidence-informed-based structures for the corresponding variables. These statistical distributions can be easily managed by the Monte Carlo simulation engine included in the DSS.

The designed policy can be considered positive/neutral for balancing inpatient and outpatient MH care. Nevertheless, selected areas with a negative impact will draw more attention. RTE, stability and entropy indicators highlight ecosystem performance and allow decision-makers to compare different alternatives. The underlying causal model identified by experts can be formalized as a Bayesian network that makes policy assessment possible. The expected outputs must be validated once the designed interventions become real, but unexpected circumstances may have a relevant impact on ecosystem performance (e.g., the COVID-19 pandemic).

Stability is very important in guiding decision-making because relevant variations in ecosystem performance scores could suddenly occur as a response to randomness. DSSs are especially useful when Monte Carlo simulation engines are integrated into their structure.

When the entropy is high, decision-makers have designed their policies specifically to cope with the target area's needs. This management style denotes flexibility but can also make it difficult to maintain a global perspective.

An improvement in the outpatient workforce increases both ecosystem performance (RTE) and stability and slightly decreases entropy.

By combining causal reasoning and statistical methods, decision makers can obtain a deep view of both pre-implementing and post-implementing situations. The first one approximates the reality and the second shows a potentially reasonable perspective of the policy impact on the reality. Both situations are always assessed under a theoretical paradigm used to interpret data values in terms of appropriateness level. Here the Balanced of care model has been used.

According to the obtained results, several causal relationships can be highlighted for guiding MH planning in the Gipuzkoa MH ecosystem: the visits on outpatient care services (community mental health centres) depends on the number of MH professionals, e.g., psychiatrists, nurses and psychologists. In addition, the length of stay in acute hospital care (inpatient care) depends on the availability of mental health centres and the number of health professionals. Finally, readmissions in inpatient care, which is known as revolving doors phenomena, depends on the availability and professionals of mental health care centres and previous stays at inpatient care. Knowing the causal levers, it is possible to act directly to the causes in order to potentially produce de appropriate results considering the uncertainty: to provide a more balanced and integrated MH care provision in the community.

## Acknowledgments

We would like to thank the Mental Health Network of Gipuzkoa, especially Álvaro Iruin and Andrea Gabilondo, for providing data and supporting this study. We also acknowledge the Mental Health Network of Bizkaia, especially Carlos Pereira and José Juan Uriarte, for supporting this research.

## Author Contributions

**Conceptualization:** Nerea Almeda, Carlos R. Garcia-Alonso.

**Data curation:** Nerea Almeda, Carlos R. Garcia-Alonso, Mencia R. Gutierrez-Colosia, Jose A. Salinas-Perez, Alvaro Iruin-Sanz.

**Formal analysis:** Nerea Almeda, Carlos R. Garcia-Alonso.

**Funding acquisition:** Carlos R. Garcia-Alonso.

**Investigation:** Nerea Almeda, Carlos R. Garcia-Alonso.

**Methodology:** Nerea Almeda, Carlos R. Garcia-Alonso.

**Software:** Carlos R. Garcia-Alonso.

**Supervision:** Carlos R. Garcia-Alonso.

**Validation:** Nerea Almeda, Carlos R. Garcia-Alonso.

**Writing – original draft:** Nerea Almeda, Carlos R. Garcia-Alonso.

**Writing – review & editing:** Nerea Almeda, Carlos R. Garcia-Alonso, Mencia R. Gutierrez-Colosia, Jose A. Salinas-Perez, Alvaro Iruin-Sanz, Luis Salvador-Carulla.

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
