## [Decision Letter · Decision Letter 0]

26 Jul 2021

PONE-D-21-17785

Modelling the balance of care: impact of an evidence-informed policy on a mental health ecosystem

PLOS ONE

Dear Dr. Salinas-Perez,

Thank you for submitting your manuscript to PLOS ONE. After careful consideration, we feel that it has merit but does not fully meet PLOS ONE’s publication criteria as it currently stands. Therefore, we invite you to submit a revised version of the manuscript that addresses the points raised during the review process.

We recommend that it should be revised taking into account the changes requested by the reviewers. Since the requested changes include **Minor Revision**, the revised manuscript will undergo the next round of review by the same reviewers.

We look forward to receiving your revised manuscript.

Kind regards,

Afnizanfaizal Abdullah, PhD. (Computer Science)

Academic Editor

PLOS ONE

Journal Requirements:

Additional Editor Comments:

Reviewers' comments:

Reviewer's Responses to Questions

**Comments to the Author**

1. Is the manuscript technically sound, and do the data support the conclusions?

Reviewer #1: Yes

Reviewer #2: Yes

2. Has the statistical analysis been performed appropriately and rigorously? 

Reviewer #1: Yes

Reviewer #2: I Don't Know

3. Have the authors made all data underlying the findings in their manuscript fully available?

Reviewer #1: Yes

Reviewer #2: Yes

4. Is the manuscript presented in an intelligible fashion and written in standard English?

Reviewer #1: Yes

Reviewer #2: Yes

5. Review Comments to the Author

Reviewer #1: This is an interesting maunscript on Causal relationships between inpatient and outpatient care variables. though the use of multivariate linear regression (generalized least squares) and Bayesian network and regression results, a decision support system is examined, combining data envelopment analysis, Monte Carlo simulation and fuzzy inference.

It is a very interesting paper and I congratulate the authors on it. While this type of strategy is widely used in other fields, it is less common in the field of mental health. This is an added value to the work.

As the authors are well aware, the term casual relationship in this field is a delicate one. My suggestion is a minor one, but I think the authors should be more careful in explaining this term to avoid confusion about it and the interpretation of the results.

Reviewer #2: 1. Write sentences straight to the point, avoid usage of too much brackets, for example Basque Country (Spain). Use Basque Country Spain, or Basque, Spain. Other example..(15 in this study) in the Section; Scenarios.

2. Ensure that the acronyms used are written in full for the first time since the readership of papers in Plos One journals come from various disciplines.

3.Methodology should be explained in more detail for example that the Gipuzkoa has a number of communities.

Also there should be explanation on the pre and post of the policy implementation. This is only mentioned later in the last paragraph of page 21 which discuss on the results. This is important to reflect the discussion in which the authors mentioned that “This model has been used to assess the potential impact of

a specific policy designed to balance inpatient and community-based care provision”

4. Page 16 -DESDE-LTC codes need to be explained more in the earlier part of the article

5. Who were the experts? What is the criteria that defines them experts?

How many experts were involved?

6. Figure three does not reflect Relationship between and .

7. Discussion can be enhanced more.

8. Conclusion can be improved by reflecting back to the objectives of the study.

6. PLOS authors have the option to publish the peer review history of their article (what does this mean?). If published, this will include your full peer review and any attached files.

Reviewer #1: No

Reviewer #2: No

---

## [Author Response · Author response to Decision Letter 0]

3 Sep 2021

Response to Reviewers

Journal Requirements

We note that the grant information you provided in the ‘Funding Information’ and ‘Financial Disclosure’ sections do not match. When you resubmit, please ensure that you provide the correct grant numbers for the awards you received for your study in the ‘Funding Information’ section.

Many thanks, we have not been able to correct this mistake in the submission system. The final financial disclosure sections is:

Financial support for this study was provided in part by a grant from the Carlos III Health Institute (PI18/01521), and the Regional Government of Andalusia (PY18-RE-0022), with European Union FEDER teams. The funders had no role in study design, data collection and analysis, decision to publish, or preparation of the manuscript.

We note that you have stated that you will provide repository information for your data at acceptance. Should your manuscript be accepted for publication, we will hold it until you provide the relevant accession numbers or DOIs necessary to access your data. If you wish to make changes to your Data Availability statement, please describe these changes in your cover letter and we will update your Data Availability statement to reflect the information you provide.

We have uploaded the dataset to Dryad Digital Repository, and it will be available for the public once the article has been published. In addition, Dryad has provided a temporal link for reviewers so they can access to the dataset. This is the link: 

https://datadryad.org/stash/share/eds2HxYMKOcALM8zfO7_bZqs5T6vKspmW8rUlOiPKJc

Nevertheless, in the text, we should maintain the original DOI for public access because the reviewers link is temporal and only for revision process. 

Additional Editor Comments

Reviewers' comments

Reviewer #1

This is an interesting manuscript on Causal relationships between inpatient and outpatient care variables. though the use of multivariate linear regression (generalized least squares) and Bayesian network and regression results, a decision support system is examined, combining data envelopment analysis, Monte Carlo simulation and fuzzy inference. It is a very interesting paper and I congratulate the authors on it. While this type of strategy is widely used in other fields, it is less common in the field of mental health. This is an added value to the work. As the authors are well aware, the term casual relationship in this field is a delicate one. My suggestion is a minor one, but I think the authors should be more careful in explaining this term to avoid confusion about it and the interpretation of the results.

Thank you very much for your kind comments. We have included additional explanations about the meaning of a causal relationship as well as the way of designing them and how to manage them in order to draw practical conclusions. For example:

In the introduction …

Any real intervention involving these elements will have a real impact, but a priori, it is impossible to know exactly what that impact can be. Any impact assessment needs to estimate reasonable potential consequences (13). In this potential underlying causal model, variable relationships should be defined by experts because they can identify potential “causes” and “effects”, as well as different “causal levels” that can define which variables can be “effects” and then “causes” of others. A causal model can be represented by a graph by linking “causes” and “effects” by, usually unidirectional, arrows. This structure is relatively easy to understand and is a powerful tool to guide statistical analysis. In this study, a Bayesian network (Direct Acyclic Graph) is selected to design the causal model.

In Techniques …

The first step was to design a set of expert-driven Bayesian network prototypes. These graph-based structures define potential causal relationships between variables based on expert knowledge. Here the experts have to decide which variable is a “cause” and which variable is an “effect” according to his/her knowledge. The resulting diagram (usually a Direct Acyclic Graph) represents an individual o collective (more appropriate) proposal that can, or not, be statistically confirmed by data. Without this underlying model it is very difficult to carry out any exploratory and confirmatory analysis or, even worse, to draw conclusions from blind regression or classification techniques. Proposed graphs must be checked iteratively, Expert-based Cooperative Analysis (24), by the panel because variable selection and dependency levels are usually difficult to define.

and …

The proposed Bayesian network defines the independent and dependent variables of each potential causal relationship. These relationships were iteratively modified and improved according to the regression results. Expert panel defined the first model and checked both the statistical results and the proposed changes in the Bayesian network until they agreed with the results.

The best Bayesian networks and their corresponding regression models were used to estimate the statistical ranges or confidence intervals for the dependent variable (potential effect, consequence, or output) given the independent variable values. These ranges, adjusted to delimited statistical distributions (triangular, trapezoidal and gamma), were managed by a Monte Carlo simulation engine for intervention assessment.

In Conclusions …

It is possible to design expert-based Bayesian networks independently from the number of observations. This approach can be more realistic for causality formalization (15), but it is not enough. Data will be necessary to confirm, if they exist, the mathematical structure of the relationship or emerging behaviour. This process must be designed in an iterative way to explore different algebraic structures (linear, product units and/or exponential) automatically to find the best resulting model (regression analysis). Therefore, statistical analysis is used to discover hidden relationships that are difficult to highlight by experts. These new relationships must be confirmed by the experts and then included in the Bayesian network, that is, the result of a combination of expert knowledge and statistical-based results. Results have shown that is possible to identify the causal relationships according inpatient and outpatient care according to the expert knowledge and from a statistical point of view. Nevertheless, it is important to note that the resulting Bayesian network is specific for the ecosystem under study. Each new situation has to be analysed according the existing, local, knowledge but the methodology is completely replicable.

Once the potential causal relationships, and their levels, are confirmed for the situation, results must be integrated for the assessment of the policy impact. This analysis have to understand that reality is not under certainty but completely uncertain. This means that each specific data (for example, the frequentation of Tolosa -catchment area) is a statistical distribution, the problem is which one. The best statistical models, specifically their confidence intervals, are used to identify the statistical distributions which are evidence-informed-based structures for the corresponding variables. These statistical distributions can be easily managed by the Monte Carlo simulation engine included in the DSS.

and …

By combining causal reasoning and statistical methods, decision makers can obtain a deep view of both pre-implementing and post-implementing situations. The first one approximates the reality and the second shows a potentially reasonable perspective of the policy impact on the reality. Both situations are always assessed under a theoretical paradigm used to interpret data values in terms of appropriateness level. Here the Balanced of care model has been used. 

According to the obtained results, several causal relationships can be highlighted for guiding MH planning in the Gipuzkoa MH ecosystem: the visits on outpatient care services (community mental health centres) depends on the number of MH professionals, e.g., psychiatrists, nurses and psychologists. In addition, the length of stay in acute hospital care (inpatient care) depends on the availability of mental health centres and the number of health professionals. Finally, readmissions in inpatient care, which is known as revolving doors phenomena, depends on the availability and professionals of mental health care centres and previous stays at inpatient care. Knowing the causal levers, it is possible to act directly to the causes in order to potentially produce de appropriate results considering the uncertainty: to provide a more balanced and integrated MH care provision in the community.

Reviewer #2

Thank you very much for your kind comments.

1. Write sentences straight to the point, avoid usage of too much brackets, for example Basque Country (Spain). Use Basque Country Spain, or Basque, Spain. Other example..(15 in this study) in the Section; Scenarios.

Many thanks, we have clarified that the Basque Country is an autonomous community located in Spain. In addition, we have followed your recommendations to make the manuscript easier to understand by readers. Every change has been highlighted in yellow. 

2. Ensure that the acronyms used are written in full for the first time since the readership of papers in Plos One journals come from various disciplines.

Many thanks, we have carried out this change in all the cases.

3.Methodology should be explained in more detail for example that the Gipuzkoa has a number of communities. Also there should be explanation on the pre and post of the policy implementation. This is only mentioned later in the last paragraph of page 21 which discuss on the results. This is important to reflect the discussion in which the authors mentioned that “This model has been used to assess the potential impact of a specific policy designed to balance inpatient and community-based care provision”

Thanks again. The structural description of Gipuzkoa has been included in the Samples and Variables section as follows:

Gipuzkoa has a population of 640,635 adults older than 17 years of age in 2015. It is one of the three historic territories of Basque Country autonomous community in Spain. The Department of Health in each historic territory has total governance capacity and centralizes healthcare management and provision (25). The MH ecosystem of Gipuzkoa is structured in 13 catchment areas, which are considered the decision-making units (DMUs) for policy assessment. Each catchment area of Gipuzkoa corresponds to a community MH centre. A single acute MH hospital unit provides care to all the DMUs (16). The 13 catchment areas are: Alto Deba-Arrasate, Amara, Andoain, Azpeitia, Beasain, Eguia, Eibar, Irun, Ondarreta, Renteria, Tolosa, Zarautz and Zumarraga.

Some additional explanations on the pre and post-implementation of the policy have been included in the paper.

In the Introduction:

Any real intervention involving these elements will have a real impact on the pre-implementing situation (reality), but a priori, it is impossible to know exactly what that impact can be, the post-implementing situation. Any impact assessment needs to estimate reasonable potential consequences (13) by defining a “reasonable” potential post-implementing situation according to expert-based expectations. In this potential underlying causal model, variable relationships should be defined by experts because they can identify potential “causes” and “effects”, as well as different “causal levels” that can define which variables can be “effects” and then “causes” of others. A causal model can be represented by a graph by linking “causes” and “effects” by, usually unidirectional, arrows. This structure is relatively easy to understand and is a powerful tool to guide statistical analysis. In this study, a Bayesian network (Direct Acyclic Graph) is selected to design the causal model.

In the Decisions (input, time t) section:

The objective is to rebalance care provision in the pre-implementing policy situation (reality) by increasing the number of high qualified professionals in outpatient care.

In the Discussion section:

It is worthy to note that they consider the DSS a useful tool for supporting and guiding policymaking, nevertheless, the analysis of the concordance between the reality (pre-implementing situation) and the predictions (potential post-implementing situation) is critical to validate the model.

4. Page 16 -DESDE-LTC codes need to be explained more in the earlier part of the article

We have described DESDE-LTC in detail:

The Description and Evaluation of Care Services and Directories for Long Term Care (DESDE-LTC) tool was used for standardized care provision (20–22). DESDE-LTC is a classification system for coding care teams and service availability, allowing international comparisons across different jurisdictions. In this study, Care Teams or Basic Stable Inputs of Care provide the following main types of care: Residential (“R” DESDE-LTC code) and Outpatient care (“O” DESDE-LTC code).

We also clarify what specific code are used and their description in de policy section:

Decision-makers expected an increase in the number of visits in outpatient care facilities, specifically the non-mobile and non-acute services with codes O8-O10 according to DESDE-LTC, and decreases in both the length of stay and the number of readmissions in inpatient care in the catchment areas, specifically the services that provide acute hospital medium intensity care with code R2 according to DESDE-LTC.

5. Who were the experts? What is the criteria that defines them experts?

How many experts were involved?

Many thanks, we have explained that question in more detail: 

It is worthy to highlight that those experts who participated in this panel were senior managers, psychiatrists, clinical psychologists, nurses and researchers, who had experience in planning and managing MH services and worked in the MH Network of Gipuzkoa and Bizkaia (another historic territory in the Basque Country, Spain). Altogether 15 experts participated in designing the scenarios and the policy to evaluate.

6. Figure three does not reflect Relationship …

Thanks again. We have clarified this issue in the Frequentation (visits) in outpatient care (time t+t1) section:

A strong exponential relationship (significance level 0.05, R2=0.9221, F=130.22) among frequentation (number of visits in community mental health centres in thousands) and the corresponding multiplication between the target population by the number of professionals providing outpatient care [3] (the resulting multiplication can be considered a composite indicator represented in Figure 2) was also found:

7. Discussion can be enhanced more.

We have enhanced the discussion and added the following paragraphs:

… It is worthy to note that they consider the DSS a useful tool for supporting and guiding policymaking, nevertheless, the analysis of the concordance between the reality and the predictions is critical to validate the model.

It is worthy to note that he relationship between outpatient care provision and general inpatient care improvement is acknowledged (29,30). In the selected policy developed in this research, the effects on the visits in outpatient care services occur in weeks or months, but those corresponding to inpatient care services in terms of length of stay, discharges and readmissions are expected to have a significant delay, taking place in months or years. 

Results obtained by Weiss et al. (2012) (31) showed the importance of integrating emergency departments in a MH system that provide aftercare to promote the transition of patients from inpatient care. Findings from Hansagi et al. (2000) (32) evidenced that users who visited actively emergency department also heavily use general health care services. These facts result interesting taking into consideration the health crisis caused by Covid-19 pandemic. The change of paradigm that we are living has forced the development of new ways of treatment in order to avoid physical contact and spread the virus. In this sense, telepsychiatry has played a crucial role in providing MH care, which has rapidly evolved because of the pandemic (33). This type of care presents advantages such as it is easier to provide continuity of care, to triage emergency patients and to reduce economic costs in providing outpatient care (34). Although the pandemic has triggered the implementation of telepsychiatry, previous studies focused on the future of psychiatry, highlighting the importance of technology to transform the traditional MH care provision (35–37). 

… Previous work on the assessment of a policy intervention in Bizkaia (Basque Country) showed that, after shifting workforce capacity from an outpatient to day care service, the global performance of the MH system increased (17). 

Future lines of work could be focused on improving outputs used for assessing relative technical efficiency by including telephonic sessions in outpatient care, which have been increased after the covid pandemic.

8. Conclusion can be improved by reflecting back to the objectives of the study.

Many thanks, we have carried out your recommendations and added the following paragraphs:

The objectives of this article were: (i) to design a formal causal model (Bayesian network prototype) based on expert knowledge and the balance of care model linking inpatient and outpatient MH care, (ii) to identify, if they exist, statistical relationships between selected variables, and finally (iii) to assess the potential impact of a specific policy enhancing the inpatient-outpatient MH care balance on the MH system of Gipuzkoa. 

Results have shown that is possible to identify the causal relationships according inpatient and outpatient care according to the expert knowledge and from a statistical point of view. Nevertheless, it is important to note that the resulting Bayesian network is specific for the ecosystem under study. Each new situation has to be analysed according the existing, local, knowledge but the methodology is completely replicable.

Once the potential causal relationships, and their levels, are confirmed for the situation, results must be integrated for the assessment of the policy impact. This analysis have to understand that reality is not under certainty but completely uncertain. This means that each specific data (for example, the frequentation of Tolosa -catchment area) is a statistical distribution, the problem is which one.

By combining causal reasoning and statistical methods, decision makers can obtain a deep view of both pre-implementing and post-implementing situations. The first one approximates the reality and the second shows a potentially reasonable perspective of the policy impact on the reality. Both situations are always assessed under a theoretical paradigm used to interpret data values in terms of appropriateness level. Here the Balanced of care model has been used. 

According to the obtained results, several causal relationships can be highlighted for guiding MH planning in the Gipuzkoa MH ecosystem: the visits on outpatient care services (community mental health centres) depends on the number of MH professionals, e.g., psychiatrists, nurses and psychologists. In addition, the length of stay in acute hospital care (inpatient care) depends on the availability of mental health centres and the number of health professionals. Finally, readmissions in inpatient care, which is known as revolving doors phenomena, depends on the availability and professionals of mental health care centres and previous stays at inpatient care. Knowing the causal levers, it is possible to act directly to the causes in order to potentially produce de appropriate results considering the uncertainty: to provide a more balanced and integrated MH care provision in the community.

Many thanks, we have uploaded the figures to Preflight Analysis and Conversion Engine (PACE) digital diagnostic tool and, according to this, they meet the requirements.

---

## [Decision Letter · Decision Letter 1]

7 Dec 2021

Modelling the balance of care: impact of an evidence-informed policy on a mental health ecosystem

PONE-D-21-17785R1

Dear Dr. Salinas-Perez,

We’re pleased to inform you that your manuscript has been judged scientifically suitable for publication and will be formally accepted for publication once it meets all outstanding technical requirements.

Kind regards,

Afnizanfaizal Abdullah, PhD. (Computer Science)

Academic Editor

PLOS ONE

Additional Editor Comments (optional):

Reviewers' comments:

Reviewer's Responses to Questions

**Comments to the Author**

1. If the authors have adequately addressed your comments raised in a previous round of review and you feel that this manuscript is now acceptable for publication, you may indicate that here to bypass the “Comments to the Author” section, enter your conflict of interest statement in the “Confidential to Editor” section, and submit your "Accept" recommendation.

Reviewer #1: All comments have been addressed

2. Is the manuscript technically sound, and do the data support the conclusions?

Reviewer #1: Yes

3. Has the statistical analysis been performed appropriately and rigorously? 

Reviewer #1: Yes

4. Have the authors made all data underlying the findings in their manuscript fully available?

Reviewer #1: Yes

5. Is the manuscript presented in an intelligible fashion and written in standard English?

Reviewer #1: Yes

6. Review Comments to the Author

Reviewer #1: Thank you for addressing my comments. I think It is ready to be accepted as It could be a good contribución to the fieles of interes for the readership of the journal

7. PLOS authors have the option to publish the peer review history of their article (what does this mean?). If published, this will include your full peer review and any attached files.

Reviewer #1: No

---

## [Editor Report · Acceptance letter]

23 Dec 2021

PONE-D-21-17785R1 

Modelling the balance of care: impact of an evidence-informed policy on a mental health ecosystem. 

Dear Dr. Salinas-Perez:

I'm pleased to inform you that your manuscript has been deemed suitable for publication in PLOS ONE. Congratulations! Your manuscript is now with our production department. 

Kind regards, 

on behalf of

Dr. Afnizanfaizal Abdullah 

Academic Editor

PLOS ONE